

# Pb and Fe flow through the mire-lake complex of Skogaryd catchment - a system under anthropogenic influence

Jonas Thomsen[1], Signe Lett[1], Leif Klemedtsson[2], Delia Rösel[2], Louise Rütting[3], Katja Salomon Johansen[4], Tobias Rütting[2].

[1] Department of Geosciences and Natural Resource Management, University of Copenhagen, DK-1958 Frederiksberg, Denmark

[2] Department of Earth Sciences, University of Gothenburg, SE-40530 Gothenburg, Sweden

[3] Chair of Soil and Plant Systems, Brandenburg University of Technology Cottbus-Senftenberg, DE-03046 Cottbus, Germany

[4] Department of Biochemistry, University of Cambridge, Cambridge, United Kingdom

*Correspondance to:* Jonas Thomsen (joth@ign.ku.dk)





**Abstract**

*Sphagnum* dominated peatlands store not only vast amounts of carbon (C) but also metals derived from bedrock and anthropogenic sources. Some metals are involved in peat C stability, and some are toxic to living organisms. However, the pools of such metals and their export in relation to export of C in the form of dissolved organic C (DOC) in mire-lake complexes have received little attention. We characterized a hemiboreal mire in Sweden

previously exposed to heavy metal pollution in terms of physiochemistry, age, total C pool, lead (Pb) and iron (Fe) content. We investigated export of Fe and Pb in relation to DOC in the mire-lake complex.

We found strong and linear correlations of Fe and DOC export in streams and the export occurred in similar ratios from both the lake and mire. The export of Fe and DOC seemed to be controlled by hydrological connectivity, while the production of DOC and Fe most likely was highest at low water table depth leading to oxic conditions.

We found the Pb content exceeded threshold values for toxicity in the top layer of the peat and in stream water leaving the mire. Stream water concentrations of Pb were as for Fe and DOC, highest after drought periods. Pb isotope analysis revealed that Pb leaving the mire accumulated through anthropogenic contamination (most likely gasoline), while the Pb leaving the lake had a Pb isotopic composition from the geological background. In addition, the lake appeared to be a sink for the anthropogenic Pb leaving the mire through sedimentation. This

study suggests that peat decomposition in peatlands affected by climate change may pose a risk to not only losing stored C, but also through release of heavy metals to the local environment.



## 1 Introduction

Peatlands hold vast amounts of organic carbon (C), equivalent to 20-30% of the global pool of soil C (Gorham, 1991). Peat is formed over thousands of years and is preserved in the deep peat layers under waterlogged and anoxic conditions (Wilson et al., 2016). Large pools of metals from both atmospheric deposition and parent bedrock are bound in peat. Some metals like iron (Fe) are involved in peat stabilisation and degradation (Qin et al., 2022; Wang et al., 2017) and some metals like lead (Pb) are toxic to living organisms (Rothwell et al., 2005;

Tipping et al., 2003). Climate changes are expected to cause longer periods of drought in the boreal zone (Helbig et al., 2020) with a concurrent decrease in precipitation during summer and an increase during winter (Dore, 2005; IPCC, 2021). As a result, peatlands in the boreal zone will experience periodically lower water table, particularly in summers, and more extreme rain events (Zhong et al., 2020). These changes to peatland hydrology might expose the deeper peat to oxygen and promote C turnover and stimulate element export from peatland ecosystems (Chen

et al., 2020; Zhang et al., 2025). How the export of DOC and hydrology affect the transport of metals is unknown for most peatlands.

    *Sphagnum* mosses dominate the vegetation in boreal peatlands and are here the main contributors to peat formation. Acidic and phenolic compounds in the *Sphagnum* cell wall provide a high cation exchange capacity (CEC) and *Sphagnum* and *Sphagnum* derived peat efficiently binds positively charged ions such as metals (Clymo,

1963; Verhoeven & Liefveld, 1997; Zhao et al., 2023). As such, peatlands are efficient sinks for natural and anthropogenic derived metals (Roux et al., 2004; Shotyk et al., 1996). In addition to the recalcitrant litter of *Sphagnum*, the environmental conditions of peatland such as cold temperatures and waterlogged conditions contribute to peat accumulation and thus metal sequestration (Aerts et al., 1992; Moore & Basiliko, 2006; Thomsen et al., 2025). Distribution of metals in the depth profile from the living top layer to the deep peat depends

on the origin of the metal. Iron is one of the most abundant metals in peatlands and mainly derived from bedrock, with a relatively small contribution from wind-blown dust (Osborne et al., 2024; Steinnes et al., 2005). Therefore, Fe concentrations increase with peat depth. Many toxic heavy metals such as Pb, mercury (Hg), arsenic (As) and cadmium (Cd) are anthropogenically sourced through deposition and are generally found in the top peat layers (González & Pokrovsky, 2014; Novak et al., 2011). Peatlands sequester a larger proportion of heavy metals than

other ecosystems (McCarter et al., 2024), but to which extend metals bound in peat are mobilised and affect downstream ecosystems are not that well understood.

    Lead has been one of the most prominent heavy metals in a variety of industries (Tchounwou et al., 2012) and Pb pollution and can be traced in different depositional environments such as lake and offshore sediments as



well as peat and peatland vegetation (Bränvall et al., 2001; Deng et al., 2004; Renberg et al., 2000). The primary

recent source of Pb in mires is from anthropogenic pollution as car fuel and ship paint that increased at the start

of the industrial revolution and peaked in the 1970s (Marcantonio et al., 2000; Renberg et al., 2000). The isotopic

composition of Pb can be used to fingerprint the source (anthropogenic versus natural) of Pb and even differentiate

between types of contamination (Gmochowska et al., 2019). Lead is toxic to microorganisms, plants and humans

and accumulates in the ecosystem (Collin et al., 2022). The maximum allowable concentration in waters for Pb

as set by the European Union (EU directive, 2013) is 14 µg L$^{-1}$, in sediments 163 mg kg$^{-1}$ dry weight and in biota

110 µg kg$^{-1}$ wet weight (Sjöberg, 2016). The high cation exchange capacity CEC of *Sphagnum* results in strongly

bound Pb, which suggests efficient safekeeping of Pb in mires under stable conditions (Dinake et al., 2019).

Due to the toxic nature of Pb, its transport is important to understand, particularly in potentially Pb

polluted peatlands (Choudhury & Panda, 2005; Doelman & Haanstra, 1979; Shakya et al., 2008). High

concentrations of Pb at the top of the mire, indicate that Pb is hardly transported vertically (Kharanzhevskaya et

al., 2023; Mariussen et al., 2017; Okkenhaug et al., 2018), even under dry conditions and heavy precipitation

events (Vile et al., 1999). However, Pb in stream water from ombrotrophic peatlands has been shown to positively

correlate with DOC (Broder & Biester, 2015, 2017; Okkenhaug et al., 2018), suggesting horizontal export of Pb

could be related to peat degradation. Whether anthropogenically sourced Pb reaches toxic levels in peatlands and

if the export of such Pb increases with concurrently higher export of DOC over time is largely unknown for most

peatlands.

During the last decade, Fe has been increasingly investigated because of two proposed mechanisms

influencing peat stability. Firstly, the "iron gate" theory suggests that oxidation of Fe, which occurs during water

table decline, induces complexation between Fe and lignin-like compounds (Wang et al., 2017). This complexation

seemingly leads to resistance against mineralization (Riedel et al., 2013; Wang et al., 2017). However, the degree

to which Fe-bound peat is protected is not entirely clear and the protecting effect of Fe may be limited to static

oxic conditions (Chen et al., 2020). At the same time, oxic conditions can also promote production of DOC and

release of $CO_2$ from OM through abiotic reactions in addition to microbial breakdown and mineralization (Brouns

et al., 2014; Fenner & Freeman, 2011; Qin et al., 2022; Strack et al., 2008). Specifically, abiotic Fe-mediated

reactions occur in the presence of oxygen and DOC (Miles & Brezonik, 1981) and have been shown to be relevant

in peat (Thomsen et al., 2025) and industrial biorefineries (Peciulyte et al., 2018) leading to a substantial

contribution to $CO_2$ production through decarboxylation processes (Page et al., 2014; Trusiak et al., 2018). These

reactions depends on Fe's ability to redox cycle (Koppenol & Hider, 2019) and primarily occur at the oxic-anoxic



interface around the water table (Qin et al., 2022; Wang et al., 2022; Yu & Kuzyakov, 2021). Therefore, changing

water table level influence DOC concentrations and $CO_2$ release from peat. The reactions leading to peat

degradation might counteract the C protecting mechanism of Fe and the stability of peat might therefore depend

not only on the water table position but also on fluctuation patterns of the water table (Brouns et al., 2014; Chen

et al., 2020). Understanding how fluctuating exposure to oxygen will affect DOC and Fe export from peatlands

should therefore be considered in relation to future climate changes when assessing mire C dynamics.

105        The controls on metal export from peatlands are not well understood, but export is influenced by pH, the

binding affinity of peat, hydrological conditions and commonly, the export of metals from peatlands correlate with

DOC (Curtinrich et al., 2024; McCarter et al., 2024).  When peat is degraded via microbial decomposition, DOC

is produced. In peatlands, temperature is important for DOC production (Rosset et al., 2022), but the most

important factor is likely oxygen availability that is linked to water table depth (Fenner & Freeman, 2011; Knorr,

2013; Strack et al., 2008). Meanwhile, the transport of DOC is primarily affected by runoff and precipitation

(Clark et al., 2007; Koehler et al., 2009). Consequently, DOC concentrations are highest in the upper parts of peat

and above the water table (Clark et al., 2008; Strack et al., 2008). Export of metals, and in particular metals with

high binding affinity to DOC, such as Fe and toxic heavy metals, increases during and after drought. Extreme

precipitation events could therefore mobilise the pools of metals accumulated after increased oxygen availability

following retracting water table in peatlands.

        Here, we investigated a mire-lake complex inland from the west coast of Sweden subjected to recent

climate change and exposed to heavy metal pollution. The mire-lake complex is part of Swedish research

infrastructure for ecosystem research that provides continuous data on soil, air and water status. We wanted to

characterize the mire in terms of physiochemistry, age and total C content and evaluate the pools and export of Fe

and Pb in relation to DOC in the light of future climate.

## 2. Materials and methods

### 2.1 Site description

This study was conducted for the period 2017-2021 in Mycklemossen mire and Ersjön lake (58°21′ N, 12°10′ E;

80 m a.s.l.), which are part of the Skogaryd research catchment in the hemiboreal forest zone in southwestern

Sweden. The Skogaryd research catchment is part of the Swedish Infrastructure for Ecosystem Science (SITES).

Air temperatures range between -0.8 °C in February and 16.8 °C in July and precipitation is highest in July to

December with approximately 75 mm per month and driest January to March with approximately 54 mm per



month (Swedish Meterological and Hydrological Institute normal means for the period 1991-2020, Fig. S1).

During the survey period, the highest annual precipitation was in 2019 and 2020 with 1150 and 1100 mm annually respectively. The lowest annual precipitation was 580 mm in 2017 (https://www.fieldsites.se/SITES). Due to climate change, summers (June, July and August) at the west coast of Sweden are predicted to receive 7% less rain in June - August, 13-18% more rain in December to February and annually be 2.2 – 3.1 °C warmer by 2041-2060 in reference to preindustrial levels (IPCC, 2021).

The mire Mycklemossen is a nutrient poor fen with bog characteristics in its vegetation such as *Sphagnum* species as the peat forming plants (Rinne et al., 2020). Two streams lead away from Mycklemossen. One runs from the north into the lake Ersjön (this stream is hereafter referred to as S1) and the other runs into a neighbouring catchment. Ersjön is 0.061 km$^2$, 4.5 m deep in the deepest part and has characteristics of a mesotrophic lake. Ersjön is sourced from Mycklemossen and from a neighbouring mostly forested catchment and

drains through a stream (hereafter referred to as S6). The total catchment area of Mycklemossen is 0.595 km$^2$ and the catchment of Ersjön is 1.337 km$^2$. The catchment is in one of the densest populated areas in Sweden and east of a highway and a former industrial ship wharf. In Europe, gasoline contained Pb from 1922 until it was banned in 1998. Mycklemossen is downwind from a potential source of industrial pollution, Uddevalla ship wharf. The ship wharf actively produced ships in the years of 1946 to 1985 and used Pb containing antifouling paint. The

area can therefore be expected to have been subjected to considerable Pb pollution during a period of 70 years.

**2.2 Peat sampling**

On March 3$^{rd}$, 2021, one peat core of 4.5 m depth was sampled at five locations in three topography types: hummock, intermediate and hollow with 1-2 meters apart. This resulted in 15 peat cores in total. The five locations

were chosen to represent Mycklemossen (Fig. 1). The top 50 cm of the peat was sampled with a hand saw. The deeper peat was collected with a stainless-steel, cylinder-shaped Russian peat corer (inner dimensions: 5 cm in diameter). Just after sampling, the peat cores were sliced into 5 cm segments and frozen.





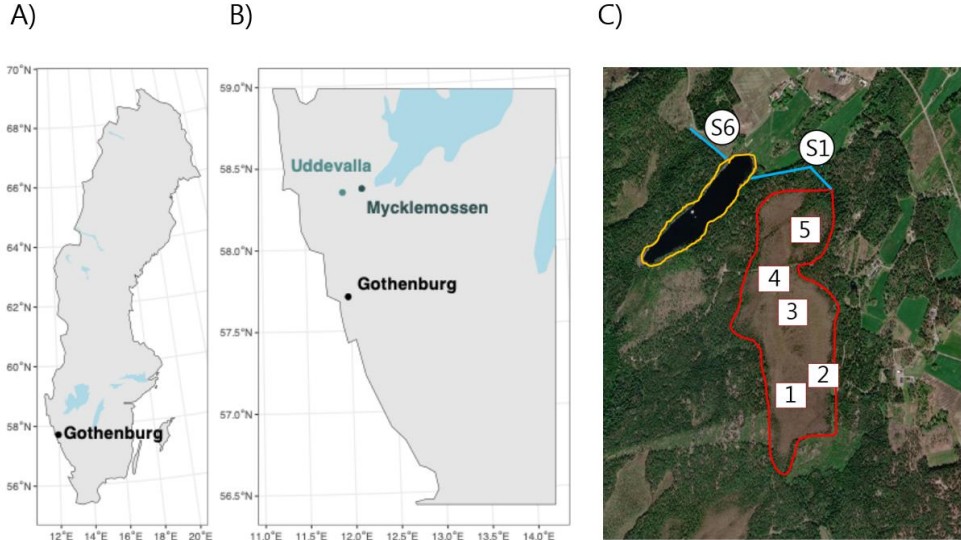


**Figure 1.** A) Map of Sweden, B) Mycklemossen mire located east of Uddevalla where a ship wharf was active until 1985, C) the mire-lake complex comprising of Mycklemossen mire (encircled in red) and lake Ersjön (encircled in yellow). Sampling points in Mycklemossen are marked as 1-5 and the stream sampling location from Mycklemossen are marked as S1 and the stream from Ersjön stream S6. Satellite photo from

https://meta.fieldsites.se/resources/stations/Skogaryd.

**2.3 Radiocarbon dating of peat**

Peat age was determined by radiocarbon dating in samples from 20-25 cm, 120-125 cm and 420-425 cm depth from hummock, intermediate and hollow from sampling location 3 (Fig. 1C). Samples were freeze dried and

homogenized prior to analysis for $^{14}$C at the Tandem laboratory (Uppsala, Sweden), using a MICADAS accurate accelerator mass spectrometry, following routine sample preparation by the tandem laboratory (Salehpour et al., 2013).

**2.4 Physical and chemical characterization of peat**

For all 15 peat cores (locations 1-5, Fig 1C) the top 5 cm of each 100 cm increment from 0-5 cm below surface to 400 cm (0-5 cm, 100-105 cm, 200-205 cm, 300-305 cm and 400-405 cm) were analysed for physical and chemical properties. pH was analysed in filtered peat pore water (Metrodam 691 pH Meter; Swiss). Subsamples of peat was oven dried at 40 °C until constant mass. Dry matter (DM) content was used to determine peat bulk



density. Soil organic matter content (SOM) was determined from loss on ignition at 550 °C for 8 hours in a muffle
furnace (Heraeus Instruments). For total C, total N, $\delta^{13}C$ and $\delta^{15}N$, dried samples were ground and weighed (5
mg) into tin capsules and analysed with a GSL elemental analyzer coupled to an isotope ratio mass spectrometer
(IRMS, Sercon 20-22, Sercon Ltd., UK).

### 2.5 Total pools of C and N in mire

To map the three-dimensional layout of the mire, bathymetry, which is based on ground penetrating radar was
performed. The radar measurements were carried out with Malå Geoscience Ramac ground penetrating radar with
a shielded 250 MHz antenna. Measurements were collected at a sampling frequency of 2600 MHz and radar depth
scans were made every 0.1 m in the different transects. For the radar measurements, the mire was divided with 4
lines from south to the north, and 10 lines east to west, forming a grid of 27 intersections, which were surveyed.
The radar measurements were calibrated against 5 soil cores taken in grid intersections using a "Russian corer".
Bathymetry data was illustrated with QGIS 3.18 and the data is available on the SITES data portal
(https://hdl.handle.net/11676.1/lmj-WqszXOi5T_XafhHU75Ps). The bathymetry data were combined with peat
bulk density and C and N content to calculate total C and in increments of 50 cm depth. Specifically, total soil
volume layer ($m^3$) at each 50 cm increment was multiplied by depth-specific peat bulk density (g $m^{-3}$) and the
average fraction of C or N.

### 2.6 Heavy metals in peat cores

The concentration of some of the highest ranked heavy metals to consider for public health (Pb, Hg, As and Cd)
(Tchounwou et al., 2012) and Fe were quantified in technical triplicates of the peat samples at 10-15cm, 15-20cm,
20-25cm, 120-125cm and 420-425 cm depth with inductively coupled plasma-mass spectrometry (ICP-MS)
(iCAP, Thermoscientific) (SI Table S1). Prior to analysis, peat samples were freeze-dried and homogenized with
a mortar and pestle. A subsample of 0.3 g was weighed into a Teflon bomb and 10 mL of concentrated nitric acid
(70 %) was added. The samples were heated in a microwave with a ramping phase of 20 minutes and a hold phase
of 25 minutes at 180 °C. After cooling, the solutions were first diluted to 50 mL with MilliQ water and further
diluted 10-fold before analysis.

### 2.7 Concentrations and export of DOC and metals in stream water



Manual grab samples from the streams were taken fortnightly during the ice-free period for chemical analysis. Water samples were filtered through 0.7 μm pore size prior to analysis. DOC was measured via spectrometry

Shimadzu TOC-VCPH. Lead (Pb) and iron (Fe) concentrations were analysed using a Agilent 8800 Triple Quadrupole ICP-MS at the Microgeochemistry Laboratory, University of Gothenburg. Samples, reference waters, and blanks were prepared in 2% $HNO_3$ with internal standards (Re, Rh, Ge) added for drift and matrix correction. The ICP-MS operated in MS/MS mode with $NO_2$ as the reaction gas. Each analyte was analyzed in four replicates of 25 sweeps at three points per peak (0.02 a.m.u. spacing). Pb was measured on mass ($^{206}$Pb) and Fe as oxide

($^{56→72}$Fe), with dwell times of 0.5 s for and 0.3 s respectively. Quantification was done in the MassHunter Workstation Software v4.4. using a six-point calibration curve (0–1000 μg/L) from the Periodic Table Mix 1 (TraceCERT®, Sigma Aldrich).

Discharge calculations are based on continuous stream level measurements using an ISCO 2110 ultrasonic flow module (Mire north) and Mjk 1400 0-1m (Ersjön outlet, S6), respectively. Location-specific

stream level discharge relation curves have been established. The data were gap-filled when necessary either via linear interpolation for shorter gaps, or by cross-correlations with nearby measurement locations within the Skogaryd catchment for longer gaps. Longer gaps usually occurred during low-flow conditions. The export of DOC and metals was calculated for each manual water sampling as the product of discharge and concentration. Linear interpolations between sampling dates were done to estimate annual export rates.


### 2.8 Pb isotope ratios in stream water

To determine the source of Pb in the mire catchment the composition of Pb isotopes was determined in stream water from both the mire and lake from the manual grab samples collected between February 2022 and May 2024 (N=25) for the mire, and between May 2022 and May 2024 (N=21) for the lake, respectively. The Pb isotopic

composition was analysed in the Microgeochemical Laboratory at the University of Gothenborg using an Agilent 8800 ICP-MS/MS coupled to an ASX-500 autosampler, a peristaltic pump, a concentric glass nebulizer (MicroMist) and a Scott double-pass quartz glass spray chamber. The autosampler was placed in an ISO 5 clean room and was connected to the ICP-MS in the adjacent laboratory with an approximately 1 m long tubing. All sample preparations were performed in the ISO 5 clean room using Milli-Q water and ultrapure nitric acid

(NORMATOM®, VWR chemicals). The ICP-MS was run in MS/MS mode with $N_2O$ as reaction gas. A comprehensive description of Pb isotopic analysis is given in the supplementary information together with summary of ICP-MS/MS settings, acquisition parameters and reference materials used (Table S2, S3, S4).



To determine the fate of anthropogenic Pb in the mire-lake complex, we applied an isotope mixing model (Eq. 1).
Assuming that Ersjön receives Pb from two different sources, water from Mycklemossen and other sources from

the surrounding forest with distinct isotopic value. Hence, the mixing model used is:

Equation 1.         $R_{Ersjön} = f_1 \times R_{Mycklemossen} + f_2 \ R_{forest}$

With R being the isotope ratio of Pb and f being the relative contribution of the two sources. The model was

applied both with the $^{206}Pb/^{207}Pb$ and $^{208}Pb/^{206}Pb$ ratios. 11 samples from a forest stream within the Skogaryd
catchment were analysed in 2023.

**2.9 Data handling and statistics**

Data were analysed and visualized in R version 4.3.3 (R Core Team, 2024). For parameters measured in peat, we

tested the effect of depth and topography type using mixed effects models with the *nlme* package (Pinheiro Bates
& R Core Team., 2023). Depth and topography type and their interaction were included as fixed, independent
predictors. For all dependent variables, except pH, depth was included as an orthogonal quadratic polynomial
term. Sampling location was included as a random factor to take the nested sampling design into account. The
relationships between annual export of DOC and annual export of Fe and Pb were tested with linear regression

including DOC and stream identity as independent variables. For the regression model with Pb as dependent
variable there was a significant interaction, and the model was run separately for the two streams.

**3 Results**

***3.1 Age, physical and chemical parameters of Mycklemossen mire***

The maximum depth of Mycklemossen was about 600 cm (Fig. 2A) and the age of the mire centre was determined
to be at least 3900 years by $^{14}C$ dating at 420-425 cm depth (Table S5). The total surface area of the mire was 0.23
$km^2$ in 2021. For the deepest part, the 550 – 600 cm depth interval, the area was only 0.0069 $km^2$ which is 0.03%
of the total area. The total C pool of Mycklemossen was 34806 ± 1633.23 t of which 18% (6340 ± 4.76 t) were in
the top 50 cm of the mire (Fig. 2B). The C and N pool declined to 1570 t ± 2.28 and 35 t ± 48 at 550-600 cm

depth, respectively.

Physiochemical parameters were measured to a depth of 400 cm. Peat bulk density (g $cm^{-3}$) significantly
decreased with depth from 0.085 ± 0.023 at the surface to 0.04 ± 0.009 at 200 cm depth and then increased to



0.064 ± 0.011 to 400 cm (p = 0.001, Fig. 3, Table S6). The SOM content was above 90% at all depths and locations

and highest in hummock. The SOM content for intermediate and hollow was lowest at the mire surface and at 400

cm depth, and highest at 200 cm depth (p = 0.04, Table S6). Peat pH was lowest at the mire surface and increased

with depth from the surface to 400 cm depth, spanning from a pH of 4.5 ± 0.18 to 5.5 ± 0.11 (Fig 3). The increase

in pH with depth was most pronounced for the hollow topography (p = 0.04, Fig 3, Table S6). Peat C content

increased slightly with depth from around 47 ± 0.8 % at the top to 51 ± 1.9 % at 400 cm, while N content decreased

from 1.27 ± 0.23 % in the surface of the mire to 0.78 ± 0.01 % in 200 cm depth and increased to 1.59 ± 0.21 % in

400 cm depth (p = 0.001, Table S6). The change in N with depth was more extreme for hummock compared to

intermediate and hollow (p = 0.04, Fig 3, Table S6). Peat $\delta^{13}$C increased from -27.5 ± 1.5‰ at the mire surface to

-26.03 ± 0.74 ‰ at 100 cm depth followed by a decrease to -27.71 ± 1.22 ‰ at 400 cm depth (p = 0.001, Fig 3,

Table S6). Peat $\delta^{15}$N decreased from -1.9 ± 1.8‰ at the surface to -2.6 ± 1.06 ‰ at 200 cm depth and then increased

to 0.06 ±1.18 ‰ at 400cm depth (p = 0.001, Fig 3, Table S6).

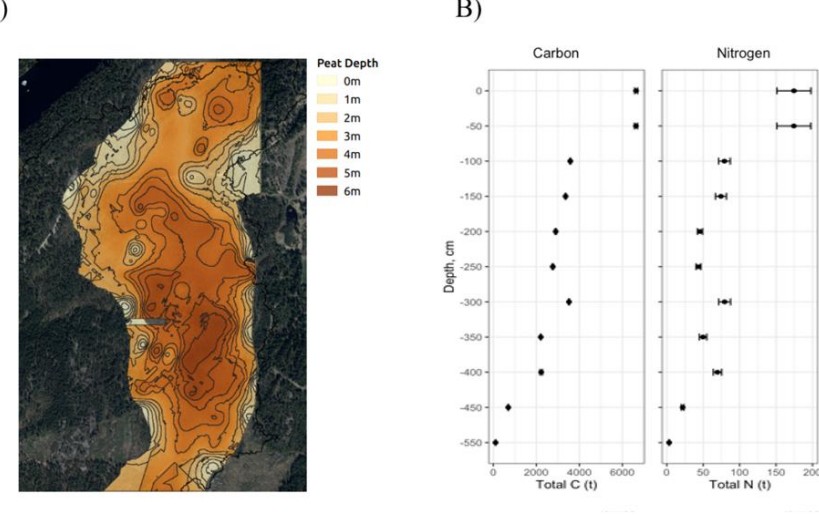


**Figure 2.** Spatial distribution of peat depth based on radar bathymetry (A) and total C and N pools in 50 cm depth

increments (B) in Mycklemossen.



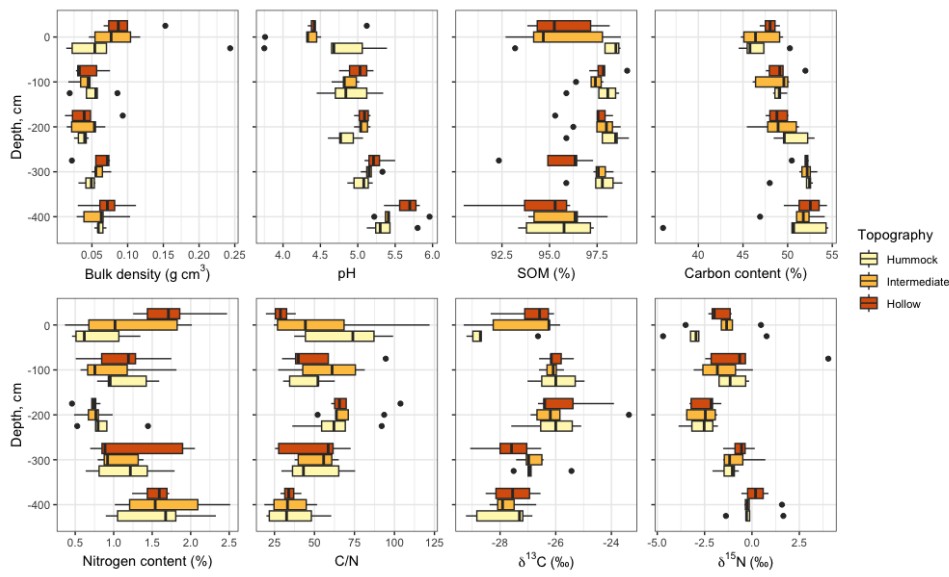


**Figure 3.** Peat properties in Mycklemossen. Peat samples in five locations and three topography types at 100 cm interval from the surface to 4 m depth. Box plots show the median value with the lower and upper quartile.

**3.2 Spatial distribution of heavy metals and Fe in Mycklemossen**

Of the four heavy metals measured, Pb was the most abundant in peat (Table S1). Peat Pb content was highest in the top of the mire and was essentially absent in peat samples at 420-425 cm depth (Fig. 4). Hummocks generally had largest Pb content, which was more than twice as high as for hollows. Specifically in hummocks, the Pb content increased from 28.26 mg kg$^{-1}$ in 15-20 cm depth to 91.8 mg kg$^{-1}$ in 25-50 cm depth. In intermediate and hollow topographies Pb content was highest in 15-20 cm: 32.21 and 64.25 mg kg$^{-1}$ and decreased to 4.41 and 0.05

mg kg$^{-1}$ at 25-50 cm depth, respectively (p = 0.02, Fig 4, Table S7). Cadmium was essentially only found in hummocks in the top 50 cm, whereas As and Cr were present throughout the peat profile of the mire (Table S1). Peat Fe concentrations at the top of the mire were between 606 and 1237 mg kg$^{-1}$ and barely changed until below 120 cm (Fig 4, Table S7).  At 420-425 cm depth, Fe content was between 7147 and 8481 mg kg$^{-1}$ and contents were similar between topography types.



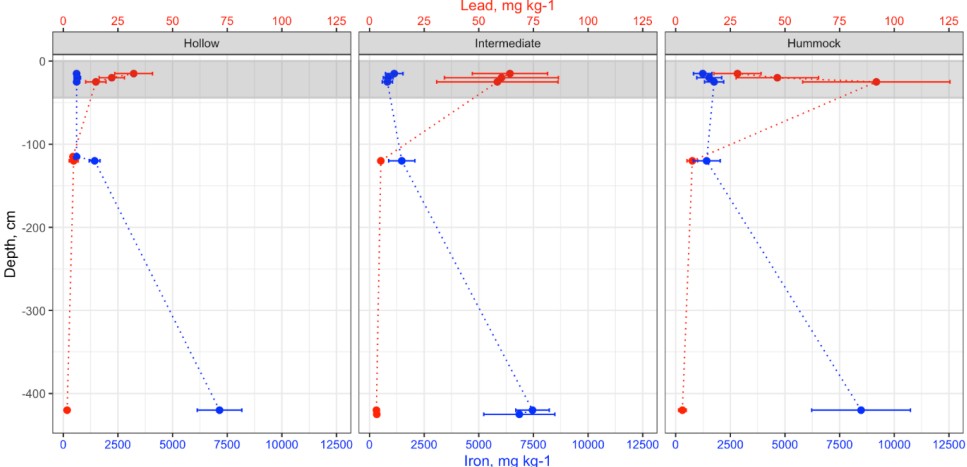


**Figure 4.** Pb (red) and Fe (blue) concentrations (mg kg$^{-1}$ DM) in peat at various depths of Mycklemossen. Points are means of 5 sampling locations and error bars are SE. Dotted lines are included for illustration only. Grey bands indicate the zone affected by water table fluctuations measured as absolute minimum and maximum water table depth during the period 2017- 2021. For statistics see Table S7.


### 3.3 Pb, Fe and DOC concentrations in export water from Mycklemossen

The discharge from Mycklemossen (S1) was lowest during summer and highest in winter. The water table concurrently decreased during summer and caused periods without discharge from S1. Stream water concentration of Pb, Fe and DOC in water from S1 were highest during summer at low discharge and water table (Fig. 5). The

highest concentrations were in general measured during summer after a drought period when discharge became measurable again. Stream water pH increased at low water table depth during summer and decreased again in winter (Fig. S2). The seasonal tendencies in Fe, Pb and DOC concentrations seemed alike, but did not follow the same pattern all the way through the study period. Lead concentrations in stream S1 were highest in 2018 at the end of the summer at the end of a drought period and was almost twice as high as in 2017 and 2019. The

concentrations of Pb in S1 during the study period varied between 0.4 µg L$^{-1}$ and 14.3 µg L$^{-1}$ (Fig. 5). Iron also peaked in 2018 at 4.6 mg L$^{-1}$, which was clearly higher than the other years, whereas DOC peaked in 2019 when the discharge increased after a drought period.



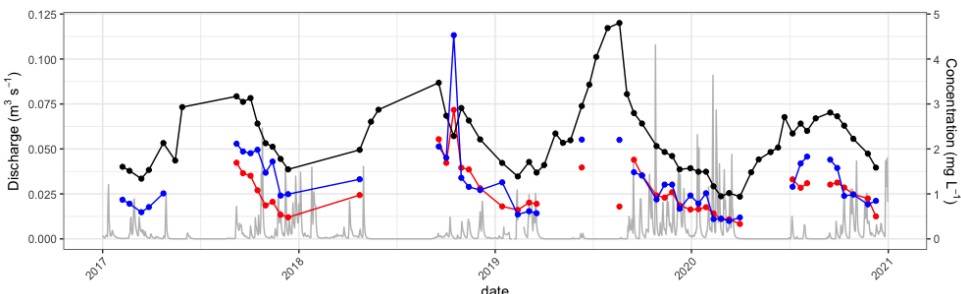

**Figure 5.** Stream discharge (grey), and water concentration of Fe (blue) and Pb x 200 (red) and C in DOC x 0.05 (black) from S1. Discharge shows daily averages of measurements made every 15 min, whereas concentrations in water were analysed from grab samples usually every 2-3 weeks during high flow and throughout 2019. Dots mark timing of sampling.

Data points for Fe and Pb were removed when discharge <0.0001 (m3 s-1).

### 3.4 Export of DOC, Fe, and Pb from Mycklemossen and Ersjön Lake

During the period 2017 – 2020 the average annual C export as DOC was 6186 kg from the north part of Mycklemossen (S1) and 12616 kg leaving Ersjön (S6). The DOC export varied considerably between years especially from Ersjön (Fig. 6). The highest DOC export from both Mycklemossen and Ersjön was in 2019, while the lowest export was in 2018. On average, annual DOC export from Ersjön in 2017 – 2020 was almost double of what it received from Mycklemossen, although this pattern was not consistent for individual years. In 2017 and 2018 DOC export from Mycklemossen and Ersjön was very similar, whereas in 2019 and 2020 export from S6 was around 3-fold higher than from S1. In 2017 –2020, the average annual Fe export was 150 kg from S1 and 325 kg from Ersjön (S6). The annual Fe export from Ersjön was therefore 215 % higher than the input from Mycklemossen. The average annual export of Pb was 0.705 kg from Mycklemossen and 0.681 kg from Ersjön.

Within streams, the inter-annual variation in Pb export followed the pattern of Fe and DOC. However, opposite to DOC and Fe, the export of Pb was only slightly higher from Mycklemossen than Ersjön. The annual export of DOC linearly correlated with annual Fe export in a similar way across the two streams (p <0.001, $R^2$ = 0.96). DOC also linearly correlated with Pb, but the correlation slopes differed between streams (p<0.001). Specifically, the annual DOC export in S6 was more than twice as high as the export from S1.





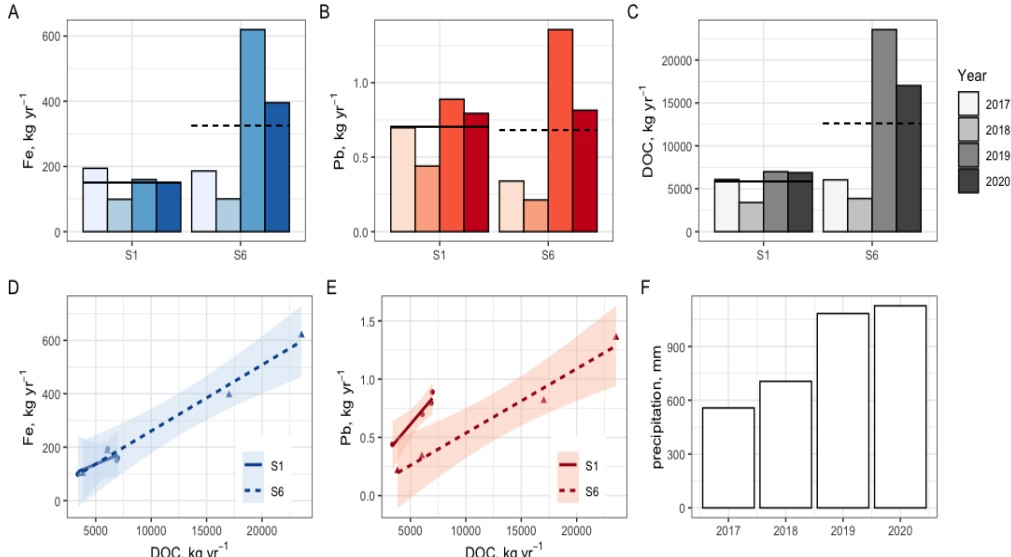

**Figure 6.** Annual export of DOC, Fe and Pb at from Mycklemossen to Ersjön lake (S1) and from Ersjön (S6) in 2017 to 2020 (A-C) and the linear relationship between export of DOC and export of Fe and Pb (D and E). Horizontal lines in bar chart are means of the years 2017-2020 in S1 (solid line) and S6 (hatched line). Note differences in scale between plots. Annual precipitation for the years 2017-2020 (F).

### 3.5 Pb isotopic composition

The Pb isotopic composition from the 57 water samples was analysed to determine whether the Pb exported from Mycklemossen and Ersjön was of anthropogenic origin (Table S8). The $^{208}Pb/^{206}Pb$ and $^{206}Pb/^{207}Pb$ composition from Mycklemossen (S1, n = 25), Ersjön (S6, n=21) and forested catchment (s12, n = 11) is plotted in figure 7, and for water leaving Mycklemossen, the $^{208}Pb/^{206}Pb$ ratio ranged between $2.106 \pm 0.004$ (2se) and $2.117 \pm 0.006$ (2se) and has a weighted mean ratio of $2.11175 \pm 0.00095$ (n=25, MSWD = 1.4). The $^{206}Pb/^{207}Pb$ ratio ranged between $1.143 \pm 0.004$ (2Se) and $1.150 \pm 0.003$ (2Se) and a weighted mean ratio of $1.1460 \pm 0.0008$ (n=25, MSWD = 1.5). For the water leaving Ersjön, the $^{208}Pb/^{206}Pb$ ratio ranged between $2.101 \pm 0.005$ (2Se) and $2.109 \pm 0.010$ (2Se) with a weighted mean ratio of $2.10690 \pm 0.0014$ (n=21, MSWD = 0.6). The $^{206}Pb/^{207}Pb$ ratio was between $1.149 \pm 0.004$ (2Se) and $1.154 \pm 0.004$ (2Se) with a weighted mean ratio of $1.1527 \pm 0.0009$ (n=21, MSWD = 0.4). For the forested neighbouring catchment, the $^{208}Pb/^{206}Pb$ ratio was between $2.062 \pm 0.009$ (2Se) and $2.088 \pm 0.008$ (2Se) with a weighted men ratio of $2.08010 \pm 0.0046$ (n=11, MSWD = 3.8). For the $^{206}Pb/^{207}Pb$





ratio, it ranged between 1.170 ± 0.005 (2Se) and 1.191 ± 0.004 (2Se) with a weighted mean of 1778 ± 0.0052

(n=11, MSWD = 9).

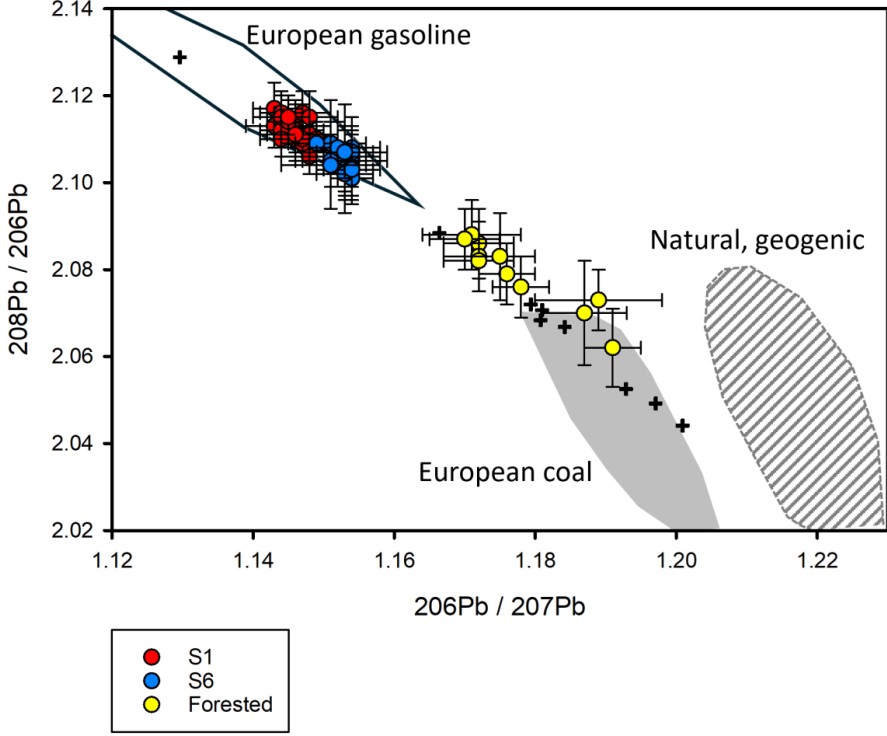

**Figure 7.** Lead isotope composition of water from S1, S6 and forest land at Skogaryd Research Catchment,

overlying an isotope plot of different Pb sources (redrawn from Komárek et al., 2008). Crosses represent the

isotopic composition of marine antifouling paint (data from Jeong et al. 2023).




**4 Discussion**

Peatlands are crucial global C stores, but subjected to anthropogenic climate change that threatens the future

stability of this C stored as peat (Zhong et al., 2020). Stored metals from natural and anthropogenic sources could

become mobilised when climate change promotes peat degradation. Here we aimed to document the size, age and

total C content of a northern European mire representing mires in the temperate and boreal transition zone and

investigate the pools and export of DOC, Fe and Pb. Specifically, we wanted to investigate the transport of Fe as

an important metal influencing peat stability and to document the Pb pollution status in the mire and a downstream

lake. We found clear indications that during warm and dry periods with higher peat degradation, metals were

released together with DOC into the downstream ecosystem. Below we discuss the context, magnitude and

consequences of these findings.

**4.1 Mycklemossen C stores and stability in relation to oxygen and Fe**

The peat in Mycklemossen was dated to 3900 years in 4 m depth and is thus likely closer to 6000 years in the

mire centre at almost 6 m deep. This is within the age range of mires in Sweden (Ehnvall et al., 2024). However,

Mycklemossen is in the younger end of the range and seems to have accumulated peat at a faster rate compared

to other Swedish mires (Andersson & Schoning, 2010; Hansson et al., 2024). The depth profile of $\delta^{13}$C and $\delta^{15}$N

partly reflects the changing age of the peat. However, we only observed minor changes (a few permill). Plant

sourced C near the surface is more $^{13}$C depleted nowadays than before the beginning of the industrial age ('the

Suess effect') (Drollinger et al., 2019). The increasingly depleted $\delta^{13}$C signature in the deepest peat layers follows

the pattern of the decreasing C to N ratio and likely reflects increasing degree of degradation with depth (Smeds

et al., 2025). Below the water table in the permanently anaerobic zone, the slightly higher $^{13}$C enrichment may

also reflect methane production activity as methanogens discriminate against $^{13}$C (Keane et al., 2021; Krull &

Retallack, 2000). The Fe content in the peat increased about 10-fold from the top to the bottom of the mire,

supporting that Fe in the Mycklemossen is primarily sourced from bedrock and groundwater and to a small extent

from deposition (Steinnes et al., 2005). 18 % of the Mycklemossen C pool was in the top 50 cm, partly because

of the large mire area at this depth interval and a higher peat bulk density in the top one meter of the mire. This

part of the mire was exposed to oxic conditions during summers, which makes a large part of the mire C

susceptible to decay. Static oxic conditions during summer could also stabilise peat due to interaction with

oxidised Fe (Chen et al., 2020), but most Fe in Mycklemossen is placed in deep anoxic peat layers, so the

stabilising effect of Fe on C might be limited. On the contrary, the C destabilising mechanism of Fe can be driven




even by small concentrations of Fe (2.3 mg g$^{-1}$ peat), which is much lower than the content in the top part of Mycklemossen (Curtinrich et al., 2022).


### 4.2 Export of DO and Fe downstream of Mycklemossen

Iron exhibited strong and linear correlations with DOC in the streams from Mycklemossen (S1) and Ersjön (S6). The average ratio between DOC and Fe of 0.03 in both S1 and S6 is the same as the median ratio value reported for Fe and DOC ratios in Swedish lakes from a national investigation (Weyhenmeyer et al., 2014). The Fe-DOC

export relationship in the mire-lake complex of Mycklemossen and Ersjön is therefore likely representative for the rest of Sweden. The strong correlation between export of Fe and DOC was expected considering that Fe transport from peatlands occurs bound to an organic ligand and also based on previous reports on co-export of Fe and DOC from (Aleshina et al., 2024; Björnerås et al., 2021; Chauhan et al., 2024).

Stream DOC reflects C loss from the mire-lake complex. DOC production is highest under oxic

conditions when the water table is low (Fenner & Freeman, 2011; Knorr, 2013; Strack et al., 2008) and at high temperatures (Rosset et al., 2022). Warm and dry years could therefore be expected to have the highest export of DOC. 2017 and especially 2018 were dry years, where the mire was exposed to oxic conditions for the longest period (Fig. S2). However, the lowest export of DOC from Mycklemossen was in 2018 and the highest DOC export was in 2019 and 2020. These last two years had up to 2-fold more precipitation compared to 2017 and 2018

(Fig. 6). In 2017 and 2018 the stream leading from Mycklemossen was dried out most of summer and therefore no export occurred in that period. The DOC production was likely highest in 2018, but lack of hydrological connectivity prevented DOC to leave the mire (Broder & Biester, 2015, 2017). The hydrologic disconnectivity is also indicated by pH values in S1 during summer much higher than the mire pH (Fig. S2). The water in S1 was during the dry summer periods not flowing and likely affected by the immediate surrounding of the stream. The

water leaving Ersjön via S6, had higher DOC export in the two wetter years, likely due to increased precipitation and discharge (Clark et al., 2007; Koehler et al., 2009; Rosset et al., 2019). In support of that, Fe, Pb and DOC in every year peaked after a drought period when the water table depth increased to a level allowing discharge measurements (Rosset et al., 2020). In general, we found that export of DOC and Fe from Mycklemossen seemed to be more affected by hydrological connectivity than warmer temperatures and oxic conditions. At the same time,

the formation of labile DOC and Fe is likely highest under oxic condition. One point this study could not address, is whether longer periods of drought, despite not increasing DOC export from Mycklemossen, increased the overall C loss through increased mineralization under oxic conditions (Fenner & Freeman, 2011).



### 4.3 Pb in Mycklemossen

In agreement with other studies, we observed highest Pb content in the top 50 cm which then dropped and reached a non-detectable content below 120 cm depth (Klaminder et al., 2006; Nieminen et al., 2002). This indicates that Pb was deposited onto Mycklemossen via wet and dry deposition. Peat in hummock was too young to date with $^{14}$C at 20-25 cm and was likely younger than the ban of lead. Hummock forming species have higher biomass production rates (Malmer & Wallén, 1999) and likely new hummock material formed on top of the polluted layer

during the last few decades since Pb was banned from gasoline and antifouling paint. In addition, hummocks are more wind exposed and therefore might be more prone for accumulating Pb compared to intermediate and hollow topographies. However, the higher Pb concentrations in hummock compared to intermediate and hollow topographies should also be seen in the light of their younger ages at 20-25 cm depth. This might lead to local differences in Pb and heavy metal toxicity on peat decomposition in peatlands depending on topography.

Pb is toxic for mosses (Choudhury & Panda, 2005) and hampers the growth of microbial communities associated with *Sphagnum fallax* (Nguyen-Viet et al., 2007). Lead contents of 80 – 300 mg kg$^{-1}$ soil can affect microbial decomposition of C and nutrient turnover (Bååth, 1989; Zheng et al., 2017) and above 70 mg kg$^{-1}$ can alter bacterial community composition (Hu et al., 2007). The Pb content of 92 mg kg$^{-1}$ in hummocks at 25-50 cm depth in Mycklemossen are thus likely to affect the microbial community and the biomass turnover rate. Most

studies that report toxic effects of Pb on microbial decomposition perform experiments in periods of weeks or months, but toxicity of Pb increases with exposure time (Connell et al., 2016). Therefore, the toxic effect of Pb could be more severe in long term *in situ* decomposition studies. Interestingly, hummock *Sphagnum* species decompose slower than hollow species under equal environmental conditions, but to which extent Pb content could play a role in their lower decomposability has not been considered (Bengtsson et al., 2016; Johnson et al.,

1990; Johnson & Damman, 1991). In Mycklemossen, we only detected contents of Pb exceeding the toxicity threshold in hummocks. Decomposition of peat was not tested in this study, but different pollution loads between topographies and local degradation dynamics might be necessary to understand Pb cycling in mires exposed to Pb deposition.

### 4.4 Pb pollution of stream waters and pb export

The concentration of Pb in stream water from Mycklemossen, was up to 10-fold higher compared to streams from pristine Siberian bogs located 200 km from anthropogenic sources of heavy metals (Kharanzhevskaya et al.,



2023), and slightly lower compared to a bog in Germany exposed to pollution from a nearby shooting range (Broder & Biester, 2015). The average Pb concentrations in stream water from Mycklemossen in the period of

2017 to 2020 was 4-fold higher than the long-term accepted values of 1.2 µg L$^{-1}$ and suggests that this stream, according to European standards, is of bad environmental quality (Sjöberg, 2016). The highest concentration of Pb in stream water from Mycklemossen was measured in late 2018 and reached 14 µg L$^{-1}$, which is the short-term maximum limit value for fresh water (Sjöberg, 2016). In stream water from Ersjön, the concentration of Pb was drastically lower (1.1±0.1 µgL$^{-1}$), and can be considered good quality for long term exposure (Sjöberg, 2016). The

14 µg L$^{-1}$ Pb concentration was measured in 2018 after a long drought period where the lowest water table depth was also measured. Just as for Fe and DOC, the highest Pb concentrations were measured after a drought period when the water table depth increased to a level allowing discharge measurements. Longer periods of drought and more extreme rain events in a future climate might therefore increase heavy metal export from polluted peatlands (Souter & Watmough, 2016; Szkokan-Emilson et al., 2013).

While twice as much Fe and DOC was exported from Ersjön through S6 compared to S1 leaving Mycklemossen, the export of Pb were similar between the two streams in the period 2017 to 2020. At the same time, about 2-fold more Pb per amount of DOC left S1 compared to S6. The explanation is probably found in how Pb in the two streams were sourced. The Pb isotope analysis revealed that Pb in S1 and S6 primarily was sourced from gasoline (Fig. 7), but the Pb isotopic signature in S6 was slightly shifted towards the signature of forest soil

which overlaps with European coal, marine anti fouling paint and natural geogenic sources. As isotope signature of marine antifouling paint falls on the mixing line of geological and gasoline Pb, its contribution unclear. Ersjön also receives water from other land-uses within its catchment, which could be dominated by natural Pb isotopes, which supplies Pb to S6 at lower concentrations than S1. Furthermore, the mixing model showed that around 33% of the Pb coming from Mycklemossen was retained in Ersjön, indicating that Ersjön acts as a sink for the Pb from

Mycklemossen. Thus, the similar Pb export from S1 and S6 is likely explained by an input from the catchment to S6 of a lower concentration than what Mycklemossen supplies, and sedimentation of Pb in Ersjön.

**Conclusion**

Here, we investigated a mire-lake complex inland from the west coast of Sweden subjected to recent climate

change and exposed to heavy metal pollution. We wanted to characterize the mire in terms of physiochemistry, age and total C content and evaluate the pools and export of Fe and Pb in relation to DOC in the light of future climate. We showed that export of Fe and DOC downstream from a nutrient poor mire located in temperate and





boreal transition zone is sensitive to climate. Furthermore, the release of Fe and DOC from the mire was most likely affected by drought conditions while the export was controlled by hydrological connectivity. The ratio

between Fe and DOC demonstrated strong linear correlations in stream water from both the mire and lake, which indicate that transport of Fe and DOC is tightly connected. Our investigation of Pb in the mire-lake complex showed that Pb pollution from gasoline had resulted in concentrations potentially toxic to microorganisms, specifically in the hummock topography and in downstream water ways. Peaks in DOC, Fe and Pb stream water concentrations occurred at increased discharge after a drought period, which showed that both Fe and Pb transport

was connected to DOC, and thus degradation of peat. As such, low water table provide conditions suitable for labile DOC production while increase in water table depth promotes DOC export. In a future climate with longer periods of drought and higher occurrence of heavy rain events, export of DOC and metals will likely increase from Mycklemossen and similar mire ecosystems. This study highlights that mires in anthropogenically exposed areas not only store C, which upon release will cause positive feedback to the climate, but also that this degradation

is linked to an environmental risk to the local environment in the form of heavy metal pollution.



**Data availability**

**Author contributions**

**Jonas Thomsen**: Conceptualization, Investigation, Methodology, Writing – original draft, Writing - review & editing. **Signe Lett**: Conceptualization, Investigation, Formal analysis, Methodology, Visualization, Writing – original draft, Writing - review & editing, Supervision. **Leif Klemedtsson**: Conceptualization, Investigation, Formal analysis, Methodology, Resources, Writing - review & editing. **Delia Rösel**: Methodology, Formal

analysis, Writing - review & editing. **Louise Rütting**: Conceptualization, Investigation, Methodology, Writing - review & editing. **Katja Salomon Johansen**: Conceptualization, Supervision, Funding acquisition, Writing - review & editing. **Tobias Rütting**: Conceptualization, Investigation, Formal analysis, Funding acquisition, Resources, Writing - review & editing.


**Competing interests**

The authors declare that they have no conflict of interest

**Acknowledgements**

Maria Shmarina, Haldor Lorimer-Olsson, David Allbrand, Josefina Carlberg is thanked for field work, peat sampling and analysis at IRMS. Martin Person is thanked for the georadar investigation.

This paper is a contribution to the Strategic Research Area "Biodiversity and Ecosystem Services in a Changing Climate" (BECC) funded by the Swedish government.


**Funding**

This work was partly funded by the Oxymist Challenge grant from the NNF (NNF20OC0059697) to KSJ and was made possible with data provided by the Swedish Infrastructure for Ecosystem Science (SITES), funded by the Swedish Research Council (VR).




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
