# Peer review of "Pb and Fe flow through the mire-lake complex of Skogaryd catchment - a system under anthropogenic influence"

_EGUsphere, 2025_

## Author Comment (AC1)

**Reviewer 3: comments and answers**

**General Comments**

The manuscript investigates the mobilization of Pb, Fe, and DOC from a mire–lake system in southern Sweden, as well as the system's carbon storage, over a four-year period marked by drought events that affected ecosystem dynamics.

The topic is relevant for understanding ecosystem responses to climate change and for the preservation of peatland–lake systems in Europe. However, the scientific conclusions are not entirely novel. As the authors acknowledge, the role of drought–rewetting cycles and hydrological connectivity in controlling DOC, Fe, and Pb export is already well established (e.g., Broder & Biester 2015, 2017; Rezanezhad et al., 2016).

The study presents a comprehensive and valuable dataset, supported by an extended four-year sampling period, which surpass typical studies based on shorter (1–2 year) campaigns. The comparison between the mire and lake compartments provides a broader perspective on the functioning of these common northern European ecosystems.

Overall, the topic is suitable for publication, but several revisions are recommended to improve the presentation and contextualization of the results. The discussion, in particular, needs a deeper and more critical analysis of the controlling processes and factors. Emphasis should also be placed on highlighting the novel aspects of this work and implications.

**Answer**

The controlling processes and factors for export of DOC, Fe and Pb will be deeper analysed in the discussion, in the next version of the manuscript.

**Specific Comments**

**Question 3.1**

The introduction would benefit from a clearer and more concise presentation of the current state of knowledge regarding Fe, DOC, and Pb export from peatlands—particularly the roles of hydrological connectivity, drought, and precipitation events (see Broder et al.). This section could be shortened by summarizing previously established processes collectively, allowing the focus to shift toward how these factors specifically affect the studied system and its long-term dynamics.

**Answer**

We agree. The paragraph starting at line 105 will be amended with the above comments in mind.

**Question 3.2**

The statement that "how the export of DOC and hydrology affect the transport of metals is unknown for most peatlands" is somewhat overstated. While some mechanistic details remain uncertain, several key processes are already considered common to peatlands (see previous comment). If previous research cannot be considered indicative for this system, it becomes difficult to reconcile this with the claim in Line 366 that the study site represents northern European mires in the temperate–boreal transition zone. Please clarified.

**Answer**

Yes, several key processes like hydrological connectivity, precipitation and drought are known drivers of DOC and metal export. The statement "how the export of DOC and hydrology affect the transport of metals is unknown for most peatlands" will be clarified in the next version of the manuscript and will be presented with more context. Our statement in the next version of the manuscript will be more along the lines of:

L50 …. In peatlands exposed to anthropogenic pollution, heavy metals will be found in the upper layers as those metals are primarily sourced from atmospheric deposition. How changes to peatland hydrology and climate affect the export of DOC and these metals is not well understood for most peatlands.

**Question 3.3**

Several trace metals (Pb, Hg, As, Cd) were analyzed in peat cores as part of previous work (Tchounwou et al., 2012). Since only Pb is discussed in the current manuscript, I recommend omitting mention of the other trace metals in both the methods and the introduction to maintain focus and clarity. If not, I recommend explaining in more detail the implication for other trace metals analyzed. For example, considering the Pb behaviour.

**Answer**

We will omit the parts about Hg, As and Cd and keep focus on Pb

**Question 3.4**

The Pb isotope dataset is valuable, covering mire, lake, and forest samples. However, its interpretation is limited. The isotope data suggest a mixing line between European gasoline, coal, and natural geogenic sources, with the lake outflow positioned between the mire and forest endmembers. This pattern likely reflects contributions from both local (forest) and upstream (mire) sources, implying that the lake catchment exerts additional influence (indicated by the forest).

In this regard, please consider:

- Topographic map illustrating the mire and lake catchments (sizes are already provided in the text) and flow directions.

**Answer**

The catchment areas have been generated based on detailed topographic maps in relation to the location of the measuring stations (out-flow) of the mire and the lake Erssjön. Thus, all the water that are measured at these locations are from these two areas. We do not think we need to add more maps to the manuscript, but will clarify this better in the text, in the next manuscript.

- Exploring implications for DOC origin: how does catchment size and type (forest, agriculture) affect lake outflow composition compared with mire outflow? Do sites S1 and S6 directly receive water from other catchment areas? Relevant literature includes Kaal et al. (2017, 2020), which highlights the contribution of forest organic matter to DOM in similar mire systems and could strengthen this interpretation.

**Answer**

Mycklemossen recieves almost all its water from rain and some from forested peatsoils, and as we discuss already, the Erssjön has a larger inflow from forest land (there is no agriculture to speak of in the catchment, just a tiny field).

- Clarifying the processes of Pb and in lake outflow. Does it primarily derive from the mire (directly). Estimating or discussing the lake's water residence time could help address this. What are the implications for the lake being a "sink" for Pb from the peatland?

**Answer**

Yes, it is primarily from the mire, which is calculated by the isotope mixing model. No previous estimates of the water residence time have been made, but given the size of the lake (6.2 ha) and the mean depth of 1.7 m (Milberg et al., 2017)) the volume would be ~100 000 m3, which gives a mean residence time of about 3 month.

- Based on isotopes and Pb. Can any kind of extrapolation be made about the role played by the mire-lake system depending on the flow (precipitation, drought period, etc.)?

**Answer**

Except for one sampling, the 206/208Pb ratio is always higher in Erssjön than Mycklemossen, proofing that the forest land almost always contributes to the lead in Erssjön. Conducting mixing model on the individual sampling dates (N=10), Mycklemossen contributes by 68 – 100% to the lead in Erssjön outflow.

Milberg, P., Törnqvist, L., Westerberg, L. M., & Bastviken, D. (2017). Temporal variations in methane emissions from emergent aquatic macrophytes in two boreonemoral lakes. AoB PLANTS, 9(4), plx029. https://doi.org/10.1093/aobpla/plx029

**Question 3.5**

Although CN ratios are presented, their implications for organic matter degradation are not discussed in sufficient depth. Given the importance of decomposition in DOM formation in peatlands, I recommend expanding on the observed CN trends and discussing how they reflect mass loss or varying degradation intensities within the cores. (See Biester et al., 2014 and Zeh eta l., 2020 for comparison between proxies for OM decomposition.

**And**

**Question 3.6**

The results for $\delta^{13}C$ and $\delta^{15}N$ show clear variations, yet their interpretation remains superficial. The authors should elaborate on how these isotopic shifts relate to organic matter degradation, plant sources, and CN ratios, and what they reveal about peat formation and transformation processes. See Zeh et al., 2020; Gandois et al., 2019.

**Answer**

Question 3.5 and 3.6 will briefly be discussed here, but will be further elobarated in the next version of the manuscript

Looking at the top part of Mycklemossen where we observe the largest differences in C/N between hollow and hummock ranging from a median value of approximately 25 to 75 for hollow and hummock respectively. (Fig. 3). First, there are differences in the vegetation between the two topographies, with the hummock being mostly dominated mainly by vascular plants (*Eriophorum vaginatum*, *Calluna vulgaris* and *Erica tetralix*) and the hollows mainly consists of different *Sphagnum* species as *S. rubellum*, *S. fallax* and *S. austinii* as well as *Rhyncospora alba* (Kelly et al., 2021). The higher median value of C/N in hummock can therefore be explained differences in the dominant vegetation, meaning the higher C/N in hummocks is likely explained by higher lignin content from vascular plants and by the presence of roots (Biester et al., 2014; Zeh et al., 2020).

The depletion of $\delta^{13}C$ in hummock contra hollow, in particular in top layer, is a common tendency and in our case, the most likely explanation is likely that hummock is dominated by vascular plants (Biester et al., 2014; Zeh et al., 2020) or a higher degree of grasses, "the Suess effect", as mentioned in the manuscript.

The increase of $\delta^{13}C$ towards 200cm depth, where hummock reach similar values to hollow and intermediate might indicate aerobic degradation of lignin in the hummock and

preservation of carbohydrates from below the water table. The lower median value of $\delta^{15}N$ in hummocks at the top of the mire contra hollow and intermediate, might be because microbial denitrification have removed light N isotope components (Biester et al., 2014). In general, the $\delta^{15}N$ increased with depth and is likely because of an uptake of the lighter N isotopes (Zeh et al., 2020).

Biester, H., Knorr, K.-H., Schellekens, J., Basler, A., & Hermanns, Y.-M. (2014). Comparison of different methods to determine the degree of peat decomposition in peat bogs. Biogeosciences, 11(10), 2691–2707. https://doi.org/10.5194/bg-11-2691-2014

Kelly, J., Kljun, N., Eklundh, L., Klemedtsson, L., Liljebladh, B., Olsson, P.-O., Weslien, P., & Xie, X. (2021). Modelling and upscaling ecosystem respiration using thermal cameras and UAVs: Application to a peatland during and after a hot drought. Agricultural and Forest Meteorology, 300, 108330. https://doi.org/10.1016/j.agrformet.2021.108330

Zeh, L., Igel, M. T., Schellekens, J., Limpens, J., Bragazza, L., & Kalbitz, K. (2020). Vascular plants affect properties and decomposition of moss-dominated peat, particularly at elevated temperatures. Biogeosciences, 17(19), 4797–4813. https://doi.org/10.5194/bg-17-4797-2020

**Question 3.7**

The differentiation among hollows, hummocks, and intermediate positions yields interesting insights into trace metal accumulation and peatland heterogeneity.

The discussion could be strengthened by integrating findings from Pérez-Rodríguez et al. (2025), who examined degradation dynamics under aerobic versus anaerobic conditions in similar microtopographies.

Additionally, it would be helpful to clarify whether the hollow–hummock pattern is assumed to have remained consistent throughout the peatland's development. And what are the possible implications. See Nungesser (2003).

**Answer**

The most interesting results in Pérez-Rodríguez et al. (2025) in relation to our study, is the leaching of phenolic compounds from the hummock topography, which could have implications for how metals are mobilised and transported. While data explaining the degradation patterns of hollow contra hummock seems interesting, the apparent selective preservation of lignin-like compounds in hollows seems a little curious. Bryophytes do not contain S-type lignin (Weng & Chapple, 2010), as otherwise shown by the paper, so the source of lignin-like compounds measured in hollows is a little unclear. The preservation og lignin-like compounds could othererwise have positive influence on metal sequestration.

Hummocks and hollows are generally relative resilient to climate shifts and the average environmental conditions (hydrology, temeperature, species composition) over time

determines the topography, that said, hummocks and hollows are formed by the species themselves (Nungesser, 2003). The time scale for development of hummock and hollow is long compared to other ecosystems, so we would assume the topography has remained the same in Mycklemossen during the time when Pb pollution was ongoing. It could imply that Pb is better sequestered in hummocks as overall slower decomposition rate is observed compared to hollows. Also, it might be worth considereing that hummock makes up most of the Mycklemossen topography (Rinne et al., 2022) and together with the hight of hummocks is more exposed to atmospheric deposition.

Nungesser, M. K. (2003). Modelling microtopography in boreal peatlands: Hummocks and hollows. Ecological Modelling, 165(2), 175–207. https://doi.org/10.1016/S0304-3800(03)00067-X

Pérez-Rodríguez, M., Alten, A., Miler, M., & Kaal, J. (2025). Explicit microrelief-controlled decoupling of initial aerobic decay and leaching (in hummocks) and anaerobic decay (in hollows) in surface layers of a Sphagnum-dominated peatland. Journal of Analytical and Applied Pyrolysis, 192, 107295. https://doi.org/10.1016/j.jaap.2025.107295

Weng, J.-K., & Chapple, C. (2010). The origin and evolution of lignin biosynthesis. New Phytologist, 187(2), 273–285. https://doi.org/10.1111/j.1469-8137.2010.03327.x

**Question 3.8** The suggestion that Pb toxicity may inhibit microbial degradation of organic matter deserves further consideration. Where is the Pb located in the moss and moss-derived organic matter, and is this Pb likely to be bioavailable to microorganisms?

**Answer**

We got a similar question from reviewer 2, that also got a similar answer.

We are aware that total Pb content is not the same as bioavailable Pb. Toxicological risk can be estimated in many ways with different ecological risk assessments (Hoang et al., 2025). We believe estimating bioavailability is out of scope for this study, as it would require some form of bioassay (Fleming et al., 2013). However, the litterature we refer to presents Pb not as bioavailable, but as total Pb (mg/kg or equivalent unit) and still found an effect on microbial processes. Thus, we believe the same could be the case for peat soils despite peats (and *Sphganum's*) ability to bind strongly to metals.

The soil characteristics: pH and organic matter content, are likely the two most important factors for heavy metal availability, of which a low pH increase availability and high organic matter decrease availability (Hou et al., 2019). In general, Pb binds strongly to the acidic and phenolic compounds of *Sphagnum* moss and its derived organic matter through physiochemical binding. Pb is probably the least mobile heavy metal and even soluble Pb will be bound to DOC because of high affinity for organic matter  (Smieja-Król et al., 2022; Vile et al., 1999).

In a study that used *Sphagnum* moss to evaluate air polution of Pb in an urban area, showed by microscopy that Pb is most likely found on the surface of the moss, which includes the inside of the large hyaline cells, in which the degree of Pb absorbtion might be affected by pore size (Dalupang et al., 2023). In *Sphagnum* derived organic matter, Pb might also be found in physical entrapments by the physiochemical binding mentioned above.

The concentration of bioavailable Pb in hummock is therefore certainly lower than the 90 mg kg$^{-1}$, but Pb can be made available from microbial degradation and lead to accumulation in organisms over time. Our statement "The Pb content of 92 mg kg$^{-1}$in hummocks at 25-50 cm depth in Mycklemossen are therefore likely to affect the microbial community and the biomass turnover rate", we find suitable, but it will be more adequately discussed in the next version of the manuscript.

Also, the Pb concentrations for water streams will be better explained in relation to bioavailability in the next version of the manuscript including some new litterature ((González & Pokrovsky, 2014; Van Sprang et al., 2016)

Dalupang, X. P., Matias, H. A., Rivera, M. L., & Viz, J. (2023). Biomonitoring of atmospheric lead (Pb) pollutants using Sphagnum moss in Bantay, Ilocos Sur, Philippines. Philippine Journal of Science, 152(6A). https://www.ukdr.uplb.edu.ph/journal-articles/6233

Fleming, M., Tai, Y., Zhuang, P., & McBride, M. B. (2013). Extractability and bioavailability of Pb and As in historically contaminated orchard soil: Effects of compost amendments. Environmental Pollution, 177, 90–97. https://doi.org/10.1016/j.envpol.2013.02.013

González, A. G., & Pokrovsky, O. S. (2014). Metal adsorption on mosses: Toward a universal adsorption model. Journal of Colloid and Interface Science, 415, 169–178. https://doi.org/10.1016/j.jcis.2013.10.028

Hoang, H. G., Hadi, M., Nguyen, M. K., Hai Nguyen, N. S., Huy Le, P. Q., Nguyen, K. N., Tran, H.-T., & Mishra, U. (2025). Assessing heavy metal pollution levels and associated ecological risks in peatland areas in the Mekong Delta region. Environmental Research, 274, 121319. https://doi.org/10.1016/j.envres.2025.121319

Hou, S., Zheng, N., Tang, L., Ji, X., & Li, Y. (2019). Effect of soil pH and organic matter content on heavy metals availability in maize (Zea mays L.) rhizospheric soil of non-ferrous metals smelting area. Environmental Monitoring and Assessment, 191(10), 634. https://doi.org/10.1007/s10661-019-7793-5

Smieja-Król, B., Pawlyta, M., Kądziołka-Gaweł, M., & Fiałkiewicz-Kozieł, B. (2022). Formation of Zn and Pb sulfides in a redox-sensitive modern system due to high atmospheric fallout. Geochimica et Cosmochimica Acta, 318, 126–143. https://doi.org/10.1016/j.gca.2021.11.032

Van Sprang, P. A., Nys, C., Blust, R. J. P., Chowdhury, J., Gustafsson, J. P., Janssen, C. J., & De Schamphelaere, K. A. C. (2016). The derivation of effects threshold concentrations of lead for European freshwater ecosystems. Environmental Toxicology and Chemistry, 35(5), 1310–1320. https://doi.org/10.1002/etc.3262

Vile, M. A., Wieder, R. K., & Novák, M. (1999). Mobility of Pb in Sphagnum-derived peat. Biogeochemistry, 45(1), 35–52. https://doi.org/10.1023/A:1006085410886

**Question 3.9**

While the cited literature on peatlands as trace metal sinks is appropriate, the authors should also consider referencing the extensive work conducted by Bindler's and Kylander's groups on Swedish peatlands, which would provide useful regional context.

**Answer**

We will consider it, and so far, we will at least include the following work:

Kylander, M. E., Bindler, R., Cortizas, A. M., Gallagher, K., Mörth, C.-M., & Rauch, S. (2013). A novel geochemical approach to paleorecords of dust deposition and effective humidity: 8500 years of peat accumulation at Store Mosse (the "Great Bog"), Sweden. Quaternary Science Reviews, 69, 69–82. https://doi.org/10.1016/j.quascirev.2013.02.010

**Question 3.10**

The citation (González & Pokrovsky, 2014) in line 64 is not appropriate. Although the authors developed an excellent model to understand trace metal accumulation in mosses, their results are not specifically related to the peatland context.

**Answer**

Yes, that is a fair point, as (González & Pokrovsky, 2014) the result are obtained under controlled laboratory conditions and not in a peatland – the reference will not be used in line 64.

**Reviewer 3 references:**

Biester, H., Knorr, K. H., Schellekens, J., Basler, A., & Hermanns, Y. M. (2014). Comparison of different methods to determine the degree of peat decomposition in peat bogs. *Biogeosciences*, *11*(10), 2691-2707.

Gandois, L., Hoyt, A. M., Hatté, C., Jeanneau, L., Teisserenc, R., Liotaud, M., & Tananaev, N. (2019). Contribution of peatland permafrost to dissolved organic matter along a thaw gradient in North Siberia. Environmental Science & Technology, 53(24), 14165-14174.

Kaal, J., Cortizas, A. M., & Biester, H. (2017). Downstream changes in molecular composition of DOM along a headwater stream in the Harz mountains (Central Germany) as determined by FTIR, Pyrolysis-GC–MS and THM-GC–MS. *Journal of Analytical and Applied Pyrolysis*, *126*, 50-61.

Kaal, J., Plaza, C., Nierop, K. G., Pérez-Rodríguez, M., & Biester, H. (2020). Origin of dissolved organic matter in the Harz Mountains (Germany): A thermally assisted hydrolysis and methylation (THM-GC–MS) study. *Geoderma*, *378*, 114635.

Nungesser, M. K. (2003). Modelling microtopography in boreal peatlands: hummocks and hollows. Ecological Modelling, 165(2-3), 175-207.

Pérez-Rodríguez, M., Alten, A., Miler, M., & Kaal, J. (2025). Explicit microrelief-controlled decoupling of initial aerobic decay and leaching (in hummocks) and anaerobic decay (in hollows) in surface layers of a Sphagnum-dominated peatland. Journal of Analytical and Applied Pyrolysis, 192, 107295. https://doi.org/10.1016/j.jaap.2025.107295

Rezanezhad, F., Price, J. S., Quinton, W. L., Lennartz, B., Milojevic, T., & Van Cappellen, P. (2016). Structure of peat soils and implications for water storage, flow and solute transport: A review update for geochemists. Chemical Geology, 429, 75-84.

Zeh, L., Igel, M. T., Schellekens, J., Limpens, J., Bragazza, L., & Kalbitz, K. (2020). Vascular plants affect properties and decomposition of moss-dominated peat, particularly at elevated temperatures. Biogeosciences, 17(19), 4797-4813.

---

## Author Comment (AC2)

**Reviewer 1: comments and answers**

**General comments**

This manuscript, which investigates the pools and export of C, Fe, and Pb in a hemiboreal mire, is generally well written, fits well within the scope of the journal, and will likely be of interest to the audience of Biogeosciences. The findings showing increased DOC and Fe mobilization following drought periods, as well as the role of hydrological connectivity in regulating DOC and Fe export from peatlands, are consistent with previous research. Placing these results within a broader climate change context is valuable. Moreover, the inclusion of the heavy metal perspective is intriguing, and the observation that peat decomposition in peatlands affected by climate change may pose a risk not only through the loss of stored C but also via mobilization of toxic heavy metals to the surrounding environment adds an important and novel dimension to the study.

**Specific comments**

**Question 1.1**

The study objectives are not entirely clear upon reading the manuscript. The research questions are introduced without a clear structure, and the final paragraph of the introduction provides only a very general statement of the aim of the study. I recommend that the authors present the study objectives more explicitly and potentially include specific hypotheses/predictions in the introduction to better guide the reader.

**Answer**

The first paragraph in the introduction will be amended to also introduce the "anthropogenic aspect" as well as DOC, and now more explicitly addresses the overall question we investigate throughout the manuscript: How changes to peatland hydrology and climate affect the export of DOC and metals, in particular Fe and Pb and how their transport is related to DOC

1) The objective stated in the end of each paragraph in the introduction sections has been described more explicitly now.
2) For the next version of the manuscript, something along the line of: The overall aim of this study is to investigate the export of DOC in relation to Fe and Pb over a 4-year period, will be added.
   Specifically, we might add the hypothesis: Fe and Pb concentrations in stream water leaving the peatland are positively correlated with DOC, and both increase in summer due to warmer and drier conditions that enhance DOC production.

**Question 1.2**

The manuscript lacks clear definitions of the topography types and information regarding their spatial scale. It is specified that the topography types were sampled 1-2 m apart, but the typical size or extent of these structures is not described. I furthermore lack a discussion of the observed differences in the physical and chemical characteristics among different topography types.

**Answer**

Mycklemossen is composed of a mosaic of hummocks and hollows with transitions zones, called intermediate in our study, but mostly dominated by hummock (Rinne et al., 2022). The hummocks are mainly dominated by *Eriophorum vaginatum, Calluna vulgaris* and *Erica tetralix* and the hollows consists of different *Sphagnum* species, mainly *S. rubellum, S. fallax* and *S. austinii* as well as *Rhyncospora alba*, and the peat samples were sampled approximately within a square meter plots (Kelly et al., 2021).

The only parameter that was significantly different between topography types was SOM content (p=0.04), while pH, N% and C/N was significant when testing for type x depth interaction (Table S6).

SOM content is affected by decomposition contra production of new organic matter. Hummocks are generally harder for microorganisms to degrade compared to hollow species, due to a higher content of recalcitrant components, primarily non-carbohydrate compounds as lignin-like compounds and secondary metabolites (Limpens et al., 2017; Mäkilä et al., 2018; Turetsky et al., 2008). Therefore, the higher SOM in hummock compared to the other topographies, particularly in the top of the mire, is likely due to less decomposition taking place in relation to production of new biomass compared to hollow species.

The formation of secondary metabolites and more structural compounds is likely also the reason for the higher N% in hollow species, as the metabolites and more structural compunds are high in C content.

Kelly, J., Kljun, N., Eklundh, L., Klemedtsson, L., Liljebladh, B., Olsson, P.-O., Weslien, P., & Xie, X. (2021). Modelling and upscaling ecosystem respiration using thermal cameras and UAVs: Application to a peatland during and after a hot drought. Agricultural and Forest Meteorology, 300, 108330. https://doi.org/10.1016/j.agrformet.2021.108330

Limpens, J., Bohlin, E., & Nilsson, M. B. (2017). Phylogenetic or environmental control on the elemental and organo-chemical composition of Sphagnum mosses? Plant and Soil, 417(1), 69–85. https://doi.org/10.1007/s11104-017-3239-4

Mäkilä, M., Säävuori, H., Grundström, A., & Suomi, T. (2018). Sphagnum decay patterns and bog microtopography in south-eastern Finland. Mires and Peat, 21, 1–12. https://doi.org/10.19189/MaP.2017.OMB.283

Rinne, J., Łakomiec, P., Vestin, P., White, J. D., Weslien, P., Kelly, J., Kljun, N., Ström, L., & Klemedtsson, L. (2022). Spatial and temporal variation in δ13C values of methane emitted from a hemiboreal mire: Methanogenesis, methanotrophy, and hysteresis. Biogeosciences, 19(17), 4331–4349. https://doi.org/10.5194/bg-19-4331-2022

Turetsky, M. R., Crow, S. E., Evans, R. J., Vitt, D. H., & Wieder, R. K. (2008). Trade-Offs in Resource Allocation among Moss Species Control Decomposition in Boreal Peatlands. Journal of Ecology, 96(6), 1297–1305. https://www.jstor.org/stable/20143576

**Question 1.3**

The depth resolution of the measured variables, especially for the metals, is coarse. Consequently, statements such as that on line 292; "Peat Fe concentrations at the top of the mire were between 606 and 1237 mg/kg and barely changed until below 120" are problematic, as no data are available for the interval between 50 and 120 cm depth. This limitation should be acknowledged in the manuscript.

**Answer**

Yes, there will definitely be variation in concentration of metals in the peat cores we do not see with our depth resolution (quite large std. on concentrations, SI table 1), and we acknowledge this. The text will be rephrased to a more neutral language: Peat Fe concentrations at the top of the mire were between 606 and 1237 mg kg-1 and was measured to between 1434 and 1474 mg kg-1 in 120cm depth (Fig 4, Table S1).

**Question 1.4**

The introduction currently lacks a clear motivation for including the lake measurements. The objectives stated at the end of the introduction are rather general and focus solely on the mire.

Interestingly, the substantially higher export of C and Fe from the lake compared to the inflow to the lake points to other sources than the mire. This observation could be explored further in the manuscript, and the relative contribution of the mire to the overall hydrological inflow to the lake could be clarified if this data is available.

**Answer**

The lake has more than twice as large catchment area than the mire and the discharge is substantially higher. Hence, a lot of water is coming from the forested land around the lake, and this can be expected to be high in DOC (maybe also Fe, as there are e.g. Podzols in the area). That means the lake receives a lot of DOC and Fe from the forested land, but not Pb.

This corresponds with the low Pb in all other streams in the catchment (mainly forest dominated), where the mire-streams are the exception.

The relative DOC and Fe export from station 1 relative to station 6 follows the ratio of catchment area, see table below.

| | Station 1 | Station 6 | St1 / St6 |
|---|---|---|---|
| Catchment area (km2) | 0.595 | 1.337 | 0.44 |
| Annual discharge (m3 yr-1) | 136 155 ± 35 168 | 412 878 ± 181 006 | 0.33 |
| DOC export (kg yr-1) | 5834 ± 1674 | 12616 ± 9293 | 0.46 |
| Fe export (kg yr-1) | 151 ± 39 | 325 ± 232 | 0.46 |
| Pb export (kg yr-1) | 0.705 ± 0.193 | 0.681 ± 0.520 | 1.04 |
| of which from St. 1 | | 84.2 ± 2.6%
= 0.573 ± 0.438* | |

* High uncertainty due to the variability in Pb export from Station 6

**Question 1.5 – technical comments**

- Line 54: Replace "binds" with "bind". **Done**
- Line 57: Replace "peatland" with "peatlands". **Done**
- Line 68: Remove "and" before "can be traced…". **Done**
- Line 270: I cannot see that the change in N with depth was more extreme for hummock compared to intermediate and hollow. This is not obvious looking at Fig. 3. Should it be the other way around?
  **Answer**
  Yes, it should be the other way around, and "more extreme" has been changed to "less extreme". N content increases in a somewhat linear manner with depth in hummock, while for intermediate and hollow, in particular for hollow, N content decreases from the top of the mire to 200 cm depth, from where it increases to 400 cm depth.

- Line 286: Intermediate generally had the highest Pb content, although the largest concentration was found in hummock at 25-50 cm (Table S1).
  Yes, that is correct.
  **New sentence:** Intermediate generally had highest Pb content that was more than twice as high as for hollows, while the highest Pb content was measured in hummock at 25-50 cm depth

- Line 288 – 290: Make sure that the correct numbers are presented here. According to Table S1, intermediate has the Pb content of 64.25 mg/kg, and hollow that of 32.21 mg/kg, and not the other way around. Pb contents of 4.41 and 0.05 mg/kg in the 25-50 cm interval do not match with the data in Table S1, nor with Fig. 4. This has been corrected
  **New sentence:** In intermediate and hollow topographies Pb content was highest in 15-20 cm: 64.25 and 32.21 mg kg$^{-1}$ and decreased to 58.5 and 14.89 mg kg$^{-1}$ at 25-50 cm depth, respectively (p = 0.02, Fig 4, Table S7).

- Line 319: It is not clear why data points for Fe and Pb were removed when discharge was low? It would have been informative to include this data.
  **New sentence:** Data points for Fe and Pb were removed when discharge <0.0001 (m3 s-1), which occurred during summer when the water level was too low to measure discharge, making calculation of export impossible.

- Line 337: Remove "at" before "from Mycklemossen". **Done**
- Line 402: Incomplete sentence starting with "The strong correlation…" **Done**
- Line 407: I suggest adding "The year of" or something similar before 2017 to avoid beginning the sentence with a number. **Done**

**Question 1.6**
Line 390: Please elaborate on what type of interaction with Fe that stabilizes peat. Also in the same sentence, that most Fe in Mycklemossen is placed in deep anoxic peat layers does not rule out that this stabilizing effect of Fe on C is important.

**And**

**Question 1.7**
Line 392: What is the "C destabilizing mechanism of Fe". Please clarify.

**Answer**

The text from Line 390 and to the end of that paragraph has been amended with comment 1.5 and 1.6 in mind. Both the stabilizing and destabilizing reactions with Fe has been elaborated.

We suggest including the following elaborated text:

Static oxic conditions during summer could also stabilise peat OM and DOC through adsorption and complexation with Fe (Chen et al., 2020; Riedel et al., 2013; Song et al., 2022), though most Fe in Mycklemossen is placed in deep anoxic peat layers, and therefore the stabilising effect of Fe on peat might be limited. However, regardless of redox regime, the majority of total Fe in peat will interact with peat OM, and Fe-OM complexes are found in both oxic and anoxic peat (Bhattacharyya et al., 2018). Thus, the stabilising effect of Fe

might not be limited to the oxic layer. On the contrary to the C stabilizing role of Fe, under aerobic conditions, oxidative reactions catalysed by Fe can lead to production of hydroxyl radicals that can promote degradation of peat (Qin et al., 2022; Trusiak et al., 2018). These reactions might be driven even by small concentrations of Fe (between 280 - 2.300 mg Kg$^{-1}$ peat) (Curtinrich et al., 2022), which is in the range of Fe contents measured in the top part of Mycklemossen. The importance of the stabilizing interactions contra the destabilizing reactions for peatland C dynamics need further investigation.

**The only new references are the oneS below here:**

Bhattacharyya, A., Schmidt, M. P., Stavitski, E., & Martínez, C. E. (2018). Iron speciation in peats: Chemical and spectroscopic evidence for the co-occurrence of ferric and ferrous iron in organic complexes and mineral precipitates. Organic Geochemistry, 115, 124–137. https://doi.org/10.1016/j.orggeochem.2017.10.012

Song, X., Wang, P., Van Zwieten, L., Bolan, N., Wang, H., Li, X., Cheng, K., Yang, Y., Wang, M., Liu, T., & Li, F. (2022). Towards a better understanding of the role of Fe cycling in soil for carbon stabilization and degradation. Carbon Research, 1(1), 5. https://doi.org/10.1007/s44246-022-00008-2

**Question 1.8** Line 420: Could this be assessed if there are $CO_2$ flux measurements from the site?

**Answer**

At the site, $CO_2$ fluxes have been measured and those show that during the drought year of 2018, soil respiration was higher compared to "normal" years. This would support the statement that DOC production is higher under oxic conditions.

Keane, B., Toet, S., Ineson, P., Weslien, P., Stockdale, J. E., and Klemedtsson, L.: Carbon dioxide and methane flux response and recovery from drought in a hemiboreal ombrotrophic bog, Front. Earth Sci., 8, 562401, https://doi.org/10.3389/feart.2020.562401, 2021.

---

## Author Comment (AC3)

**Reviewer 2: comments and answers**

This study presents a well-designed and careful investigation of Fe-Pb-DOC dynamics in a boreal mire-lake system affected by historical anthropogenic Pb pollution. The authors integrate peat-core geochemistry, radiocarbon dating, long-term DOC-metal flux monitoring, and Pb isotopic tracing to quantify the coupling between carbon and metal export under variable hydrological regimes. The dataset is of high analytical quality, and the topic is of clear environmental significance. The main conclusions-namely, that hydrological fluctuations promote the co-export of Fe and DOC, and that anthropogenic Pb is remobilized but largely retained within the lake-are well supported by the data.

However, I have several comments that the authors may wish to consider:

**Question 2.1**

First, the reported Fe-DOC correlation ($R^2$ = 0.96) is striking but insufficiently explained. The term "hydrological connectivity" alone does not capture the underlying chemical mechanisms. Please discuss whether the observed Fe-DOC co-variation results primarily from colloidal co-transport, redox-driven Fe-organic complexation, or other processes.

**Answer**

In general, Fe in boreal rivers is primarily transported on colloidal form, for instance as Fe-DOC complexes with humic substances, and Fe as $Fe^{2+}$ or Fe(oxy)hydroxides (Heikkinen et al., 2022). The transport of these forms of Fe is primarily affected by redox conditions (oxic/anoxic) and pH, whereas oxic conditions favor co-precipitation of DOC and Fe (Riedel et al., 2013) and acidic pH (4-5) favors soluble Fe-DOC complexes, while a pH of 6 and higher favors precipitation of Fe-DOC complexes (Neubauer et al., 2013).

We believe our reported Fe-DOC relationship is mostly explained by colloidal transport of Fe-DOC complexes, with essentially all Fe bound to an organic ligans and a small proportion as Fe(oxy)hydroxides. It has been reported in a recent study that Fe in northern rivers is dominantly transported on colloidal form (between 1kDa and 0.22µm), and otherwise and to a much less extent found as "truly dissolved" (<1kDa) (Aleshina et al., 2024). Björnerås et al. (2021) investigated Fe inflow to a lake from a Mire, in the south of Sweden. Here, Fe was found bound mainly on colloidal form, but also a noteworthy proportion as Fe(oxy)hydroxides, but here the inflow water had a pH of 6.7 and above, which favors presipitation of Fe-DOC complexes. At the pH we measured (4-5), we would expect a higher fraction of soluble Fe-DOC complexes.

The asynchronous timing of Fe (2018) and DOC (2019) peaks is probably due to a higher soil respiration of DOC in 2018. Both Fe and Pb concentrations were highest in 2018, so the peat degradation was likely highest that year. In 2018, the soil respiration was 15% higher compared to "normal" years (Keane et al., 2021). 2018 was an extreme warm and dry

summer in Scandinavia, and therefore it is likely that the larger soil respiration was due to mineralization of DOC, leading the lower DOC peaks in 2018 compared to 2019, which would also explain the higher release of Pb and Fe in 2018.

The large DOC peak in 2019 is harder to explain. One possible explanation is that the stream was supplemented by DOC rich water from the catchment, as suppported by Fig. S2, that shows the the pH in the stream water in 2019 was noteably higher than 2018 (pH 5-6.5) in the summer period. The high pH is not what we expect from water coming from the mire. Furthermore, there was a much higher precipitation in 2019 compared to 2018 (see table below), that could lead to a higher runoff, rich in DOC, from the forested catchment.

| SMHI data from Vänersborg | |
| --- | --- |
| Year | Precipitation (mm) |
| 2015 | 875 |
| 2016 | 655 |
| 2017 | 708 |
| 2018 | 599 |
| 2019 | 911 |
| | |

Aleshina, A., Rusakova, M.-A., Drozdova, O. Y., Pokrovsky, O. S., & Lapitskiy, S. A. (2024). Dissolved Iron and Organic Matter in Boreal Rivers across a South–North Transect. Environments, 11(4), Article 4. https://doi.org/10.3390/environments11040065

Björnerås, C., Persson, P., Weyhenmeyer, G. A., Hammarlund, D., & Kritzberg, E. S. (2021). The lake as an iron sink—New insights on the role of iron speciation. Chemical Geology, 584, 120529. https://doi.org/10.1016/j.chemgeo.2021.120529

Heikkinen, K., Saari, M., Heino, J., Ronkanen, A.-K., Kortelainen, P., Joensuu, S., Vilmi, A., Karjalainen, S.-M., Hellsten, S., Visuri, M., & Marttila, H. (2022). Iron in boreal river catchments: Biogeochemical, ecological and management implications. Science of The Total Environment, 805, 150256. https://doi.org/10.1016/j.scitotenv.2021.150256

Keane, B., Toet, S., Ineson, P., Weslien, P., Stockdale, J. E., and Klemedtsson, L.: Carbon dioxide and methane flux response and recovery from drought in a hemiboreal ombrotrophic bog, Front. Earth Sci., 8, 562401, https://doi.org/10.3389/feart.2020.562401, 2021.

Neubauer, E., Köhler, S. J., von der Kammer, F., Laudon, H., & Hofmann, T. (2013). Effect of pH and stream order on iron and arsenic speciation in boreal catchments. Environmental Science & Technology, 47(13), 7120–7128. https://doi.org/10.1021/es401193j

Riedel, T., Zak, D., Biester, H., & Dittmar, T. (2013). Iron traps terrestrially derived dissolved organic matter at redox interfaces. Proceedings of the National Academy of Sciences, 110(25), 10101–10105. https://doi.org/10.1073/pnas.1221487110

**Question 2.2**

Second, the Pb isotope work is technically robust, and the evidence for anthropogenic Pb contamination derived from gasoline combustion is convincing. Nevertheless, I recommend that the authors provide propagated uncertainties for the isotope ratios and isotope-mixing model outputs and compare their measured isotope values with established European reference baselines (these appear to be missing from the manuscript). Moreover, please include confidence intervals for the estimated ~33% Pb retention in the lake to substantiate this quantitative conclusion.

**Answer**

First, we would like to mention that we corrected the export data for the mixing model, we unfortunately had used a wrong value in calculation before. The new results indicate that 19% of lead exported from Mycklemossen is retained in Erssjön.

For the mixing model we now used the IsoError tool (Phillips & Gregg, 2001), which provides also an uncertainty estimate. With the corrected Pb export data, 84.2 ± 2.6% of the lead exported from Station 6 is derived from Mycklemossen. With this, we can estimate how much of the Pb from Mycklemossen is retained in Erssjön, which is 19 % (retained = 0.705 – 0.681 x 84.3% = 0.132). The results are summarized in the table below, also including the other elements, area and discharge.

|  | Station 1 | Station 6 | St1 / St6 |
|---|---|---|---|
| Catchment area (km2) | 0.595 | 1.337 | 0.44 |
| Annual discharge (m3 yr-1) | 136 155 ± 35 168 | 412 878 ± 181 006 | 0.33 |
| DOC export (kg yr-1) | 5834 ± 1674 | 12616 ± 9293 | 0.46 |
| Fe export (kg yr-1) | 151 ± 39 | 325 ± 232 | 0.46 |
| Pb export (kg yr-1) | 0.705 ± 0.193 | 0.681 ± 0.520 | 1.04 |
| of which from St. 1 |  | 84.2 ± 2.6%
= 0.573 ± 0.438* |  |

* High uncertainty due to the variability in Pb export from Station 6

Regarding the European reference baselines, data on lead isotopes for agricultural soils across Europe are available (Reimann et al. 2012). For the area around Skogaryd, the map shows a 206Pb/207Pb ratio of 1.245-1-326, which is higher than the data from Mycklemossen of 1.146 (which is actually at the lowest end for Northern Europe, see table). Reimann et al. does not provide a map for 208Pb/206Pb, but the value of 2.112 for Mycklemossen is quite close to the median for N-Europe.

| | Northern Europe | | |
| --- | --- | --- | --- |
| | Minimum | Median | Maximum |
| Pb | 1.6 | 9.6 | 52 |
| $^{206}Pb/^{207}Pb$ | 1.143 | 1.258 | 1.727 |
| $^{207}Pb/^{208}Pb$ | 0.287 | 0.397 | 0.414 |
| $^{208}Pb/^{206}Pb$ | 1.477 | 2.017 | 2.702 |

Regarding propagated uncertainties: The Pb isotopic data in the provided data table include the following uncertainties propagated in quadrature: the standard error of the 10 replicates, the excess scatter of the primary reference solution NIST SRM 981 for the respective ratio from the measurement session and the uncertainty in the 204Hg correction. The ratio uncertainties from the published ratios for the primary reference material (NIST SRM 981, NIST, Cantanzaro et al. 1968) would need to be propagated to add systematic uncertainties. These uncertainties are 0.04% for the 208Pb/206Pb and 206Pb/207Pb ratios used to in Figure 7 and are thus considerably smaller than the total analytical uncertainties of approximately 0.30 % for both ratios. Thus, propagation of this systematic uncertainty would not significantly change the total uncertainties for both ratios (4[th] or 5[th] digit behind the comma for the absolute ratios).

Phillips, D.L. and J.W. Gregg (2001).  Uncertainty in source partitioning using stable isotopes. Oecologia 127: 171-179 (see also erratum - Oecologia 128: 304)

Reimann et al. (2012) A Lead and lead isotopes in agricultural soils of Europe – The continental perspective. Applied Geochemistry 27:532–542

**Question 2.3**

Third, the finding that surface peat Pb concentrations exceed ecotoxic thresholds (>90 mg kg[-1]) is both important and policy-relevant. However, the manuscript does not adequately discuss Pb speciation or its geochemical associations, which are critical for assessing Pb mobility and ecological risk.

**Answer**

Total Pb concentrations exceeded ecotoxic threshold of >90 mg kg[-1], as outlined by Sjöberg, B. (2016), in the hummock topography, and we believe the Pb originated from atmospheric deposition. Total Pb content is not the same as bioavailable Pb, but the toxicological risk can be estimated in different ways with different ecological risk assessments (Hoang et al., 2025). We believe assessing bioavailability is out of scope for this study, as it would require

some form of bioassay (Fleming et al., 2013). However, the litterature we refer to presents Pb not as bioavailable, but as total Pb (mg/kg or equivalent unit) and still found an effect on microbial processes. Thus, we believe the same could be the case for peat soils despite peats (and *Sphganum's*) ability to bind strongly to metals.

In peatlands, Pb is mostly found strongly bound to organic matter and minerals with a low available fraction (Lu et al., 2025). The soil characteristics: pH and organic matter content (including CEC), are likely the two most important factors for heavy metal availability, of which a low pH increase availability and high organic matter decrease availability (Hou et al., 2019). Under aerobic conditions, as in hummock, the most prevalent ionic form of Pb is $Pb^{2+}$ that will be found in association mainly with organic matter.

The concentration of bioavailable Pb in hummock is therefore certainly lower than the 90 mg $kg^{-1}$, but Pb can be made available from microbial degradation and lead to accumulation in organisms over time. Our statement "The Pb content of 92 mg $kg^{-1}$ in hummocks at 25-50 cm depth in Mycklemossen are therefore likely to affect the microbial community and the biomass turnover rate", we find suitable, but it will be more adequately discussed in the next version of the manuscript.

Also, the Pb concentrations for water streams will be better explained in relation to bioavailability in the next version of the manuscript including some new litterature ((González & Pokrovsky, 2014; Van Sprang et al., 2016)

Fleming, M., Tai, Y., Zhuang, P., & McBride, M. B. (2013). Extractability and bioavailability of Pb and As in historically contaminated orchard soil: Effects of compost amendments. Environmental Pollution, 177, 90–97. https://doi.org/10.1016/j.envpol.2013.02.013

Hoang, H. G., Hadi, M., Nguyen, M. K., Hai Nguyen, N. S., Huy Le, P. Q., Nguyen, K. N., Tran, H.-T., & Mishra, U. (2025). Assessing heavy metal pollution levels and associated ecological risks in peatland areas in the Mekong Delta region. Environmental Research, 274, 121319. https://doi.org/10.1016/j.envres.2025.121319

Hou, S., Zheng, N., Tang, L., Ji, X., & Li, Y. (2019). Effect of soil pH and organic matter content on heavy metals availability in maize (Zea mays L.) rhizospheric soil of non-ferrous metals smelting area. Environmental Monitoring and Assessment, 191(10), 634. https://doi.org/10.1007/s10661-019-7793-5

Lu, Z., Ning, Y., Liu, C., Song, X., Pang, Y., Li, Q., Yang, M., & Zeng, L. (2025). Geochemical Regulation of Heavy Metal Speciation in Subtropical Peatlands: A Case Study in Dajiuhu Peatland. Land, 14(6), 1256. https://doi.org/10.3390/land14061256

González, A. G., & Pokrovsky, O. S. (2014). Metal adsorption on mosses: Toward a universal adsorption model. Journal of Colloid and Interface Science, 415, 169–178. https://doi.org/10.1016/j.jcis.2013.10.028

Sjöberg, B. (2016). Miljögifter i vatten – klassificering av ytvattenstatus. Miljögifter i Vatten – Klassificering Av Ytvattenstatus.

https://www.havochvatten.se/download/18.6d9c45e9158fa37fe9f57c25/1708800059479/vagledn-miljogiftsklassning-hvmfs201319.pdf

Van Sprang, P. A., Nys, C., Blust, R. J. P., Chowdhury, J., Gustafsson, J. P., Janssen, C. J., & De Schamphelaere, K. A. C. (2016). The derivation of effects threshold concentrations of lead for European freshwater ecosystems. Environmental Toxicology and Chemistry, 35(5), 1310–1320. https://doi.org/10.1002/etc.3262

**Question 2.4**

Given that the studied mire is strongly Sphagnum-dominated, the biochemical characteristics of Sphagnum mosses likely play a central role in the observed Fe-DOC-Pb interactions. Sphagnum tissues contain abundant polyphenolic compounds and organic acids, all of which can influence Fe and Pb cycle and modulate DOC chemistry. Could the authors elaborate on how these unique Sphagnum traits might govern the tight Fe–DOC correlation and the substantial Pb retention observed in this system? For example, does the acidity and high ligand density of Sphagnum-derived organic matter affects the stability of Fe-DOC-Pb complexes? A brief discussion along these lines would substantially enhance the ecological and mechanistic relevance of the study.

**Answer**

We are not aware of the specific type of DOC that leaves the mire contra the lake, but the relationship between Fe and Pb with DOC seems a little stronger in the stream leaving the mire contra the lake.

A study that investigated chemical characteristics of DOM downstream of a bog, found that the water leaving the bog was rich in polyphenolic DOM, which binds strongly to metals like Fe and Pb and was seemingly derived mainly from *Sphagnum* (Kaal et al., 2017). A pH above 6 will increase co-precipitation of both Fe and Pb with OM, while Pb-OM complexes are more mobile under acidic conditions (pH ~4.5) (Rothwell et al., 2008). These two findings could support the tight relationship of Pb and Fe with DOC leaving the mire.

Kaal, J., Cortizas, A. M., & Biester, H. (2017). Downstream changes in molecular composition of DOM along a headwater stream in the Harz mountains (Central Germany) as determined by FTIR, Pyrolysis-GC–MS and THM-GC–MS. Journal of Analytical and Applied Pyrolysis, 126, 50–61. https://doi.org/10.1016/j.jaap.2017.06.025

Rothwell, J. J., Evans, M. G., Daniels, S. M., & Allott, T. E. H. (2008). Peat soils as a source of lead contamination to upland fluvial systems. Environmental Pollution, 153(3), 582–589. https://doi.org/10.1016/j.envpol.2007.09.009